# Optimization of Transposon Mutagenesis Methods in *Pseudomonas antarctica*

**DOI:** 10.3390/microorganisms11010118

**Published:** 2023-01-01

**Authors:** Sangha Kim, Changhan Lee

**Affiliations:** Department of Biological Sciences, Ajou University, Suwon 16499, Republic of Korea

**Keywords:** *Pseudomonas*, *Pseudomonas antarctica*, transposon, transposon insertion mutant library

## Abstract

*Pseudomonas* is a widespread genus in various host and environmental niches. *Pseudomonas* exists even in extremely cold environments such as Antarctica. *Pseudomonas antarctica* is a psychrophilic bacterium isolated from Antarctica. *P. antarctica* is also known to produce antimicrobial substances. Although *P. antarctica* can provide insight into how bacteria have adapted to low temperatures and has significant potential for developing novel antimicrobial substances, progress in genetic and molecular studies has not been achieved. Transposon mutagenesis is a useful tool to screen genes of interest in bacteria. Therefore, we attempted for the first time in *P. antarctica* to generate transposon insertion mutants using the transfer of a conjugational plasmid encoding a transposon. To increase the yield of transposon insertion mutants, we optimized the methods, in terms of temperature for conjugation, the ratio of donor and recipient during conjugation, and the concentration of antibiotics. Here, we describe the optimized methods to successfully generate transposon insertion mutants in *P. antarctica*.

## 1. Introduction

The genus *Pseudomonas* is found in a variety of habitats from hosts to environments. Certain *Pseudomonas* species are crucial pathogens of various hosts; for example, *Pseudomonas aeruginosa* and *Pseudomonas syringae* as human and plant pathogens, respectively [1,2]. Notably, *Pseudomonas* is widely found in various environments, from terrestrial to aquatic environments [3,4,5]. For instance, *Pseudomonas* exists even in extremely cold environments such as Antarctica. The existence of *Pseudomonas* spp. in Antarctica was first reported in 1976 [6]. The novel characteristics of *Pseudomonas* isolates from Antarctica have been discovered and widely applied in biotechnology. For example, the psychrophilic lifestyle and production of compounds associated with plant physiology made it possible to use the *Pseudomonas* isolate as a biofertilizer in cold environments [7]. In addition, certain *Pseudomonas* isolates from Antarctica have been used for the biodegradation of petroleum and various organic contaminants and for the production of various polyesters such as polyhydroxybutyrate and polyhydroxyalkanoate [8,9,10,11,12,13]. Among the various bacterial genera, *Pseudomonas* has a relatively large genome, suggesting that *Pseudomonas* processes a large repository of genetic information [3]. In addition, horizontal gene transfer frequently occurs in *Pseudomonas*, contributing to genetic diversity [3,14]. As described, *Pseudomonas* is closely related to human health and also has substantial potential for biotechnological applications, particularly from the species in extreme environments. However, its genetic studies have not actively been performed as in *Escherichia coli*, and also the genetic tools have not been well established.

In 2004, *Pseudomonas antarctica* was identified in aquatic habitats in Antarctica [5]. *P. antarctica* is a psychrophilic, motile with a polar flagellum, and rod-shaped bacterium [5]. The *P. antarctica* PAMC 27494 strain was isolated from a freshwater sample collected at King George Island in Antarctica and its complete genome was determined [15]. The PAMC 27494 strain contains 6,441,449 bp of the genome and two plasmids [15]. Notably, the PAMC 27494 strain exhibits killing activity toward Gram-positive bacteria by producing antimicrobial substances [15]. This antimicrobial activity of PAMC 27494 strains is thought to be due to the inclusion of a gene cluster encoding microcin B, which inhibits DNA regulation by targeting the DNA gyrase [15]. In addition, *P. antarctica* produces cell-free ice nuclei composed of lipids and lipoglycoproteins [16]. However, genetic and physiological follow-up studies on *P. antarctica* have not been adequately conducted so far.

Transposons can integrate into the genome in a random position or site-specific manner, resulting in the inactivation of the genes. Therefore, it has been widely used to generate transposon mutant libraries for genetic screening. The Tc1/Mariner transposon is widely used as a genetic tool [17]. Mariner transposases can integrate transposons in any TA dinucleotide present in the genome, providing an advantage for full genome coverage [18,19]. Transposon mutagenesis has been conducted to generate a transposon mutant library in various species of *Pseudomonas* species, such as *P. aeruginosa*, *P. putida*, *P. fluorescens*, and so on [20,21,22]. However, transposon mutagenesis has been limitedly performed in psychrophilic bacteria, suggesting the need for the optimization of transposon mutagenesis methods in psychrophilic bacteria [23,24,25]. By using transposon mutant libraries in psychrophilic bacteria, it would be possible to identify the key genetic factors for cold adaptation as well as the genes involved in the production of novel metabolites from psychrophilic bacteria. In this study, we established and also optimized the protocol of transposon insertion for the psychrophilic bacterium *P. antarctica*.

## 2. Materials and Methods

### 2.1. Strains, Plasmids, Media, and Growth Conditions

*E. coli* K-12 SM10(λpir) and *P. antarctica* PAMC 27494 strains were used in this study [15,26]. pBTK30 is a conjugational plasmid encoding an ampicillin-resistant gene and Tc1/Mariner transposon with gentamicin resistance [26]. The strain SM10(λpir) processing pBTK30 was used as a donor strain. Luria–Bertani (LB) medium was used to culture both *E. coli* and *P. antarctica* strains. For the selection after conjugation, Vogel–Bonner minimal medium (VBMM) was used with supplementing 0.2% citrate acid and gentamicin [27]. *E. coli* and *P. antarctica* were cultured at 37 °C and 20 °C, respectively, and any altered condition is described in the results and discussion section. An amount of 50 μg/mL of ampicillin was used and the concentration of gentamicin is described in the results and discussion section. After conjugation and selection, transconjugants were either streaked to isolate pure genetic single colonies or scrapped into a selective medium containing 16% of glycerol to store at −80 °C.

### 2.2. Transposon Insertion

Transposon insertion into *P. antarctica* was based on the transfer of Tc1/Mariner transposon through conjugation. SM10(λpir) pBTK30 and PAMC 27494 strains were used as donor and recipient strains, respectively. The detailed protocol for the conjugation and selection condition is described in the results and discussion section.

### 2.3. Spot Titer Assay

To evaluate the yield of transconjugants after various conditions of conjugation, a spot titer assay was performed. After conjugation, the mating mixture of donor and recipient cells was serially 10-fold diluted using a VBMM medium. An amount of 10 μL of the 10-fold diluted cells was spotted on a VBMM–agar medium containing gentamicin. As a negative control, only donor cells or only recipient cells were spotted on the plate—SM10(λpir) pBTK30 and *P. antarctica* PAMC 27494, respectively. The colonies on the plates were counted to evaluate the efficiency of conjugation.

### 2.4. PCR for the Verification of Transposon Insertion

To verify the insertion of transposon in the genome of recipient cells, PCR was performed by using primers specific to transposon sequences. Primers of the mariner transposon are Tn-F (5′-GGTTCTGGACCAGTTGCGTGAG-3′) and Tn-R (5′-TAACAGGTTGGCTGATAAGTCC-3). As a PCR control, a pair of primers binding to the *ibpA* gene in the genome were used in this study (IbpA-F: 5′-GCGCGGAATTCTTTAATCAGGAGATTTAGAGATGCATCATCATCATCATCAC GCTACTACCCTTTCGTTGGCC-3, IbpA-R: 5′-CGCTCTAGATTACTGCAAGCCAACCTGACG-3). Alternatively, a pair of primers that bind to genes with essential functions is recommended as a PCR control; for example, *dnaK*, where transposon insertion can lead to a lethal phenotype.

### 2.5. Inverse PCR and Sequencing

Inverse PCR and subsequent sequencing were performed to identify the insertion site of the transposon, as described by Green et al. [28]. In brief, gDNA from the transconjugant was isolated by using a genomic DNA extraction kit (Bioneer, Daejeon, Republic of Korea). The extracted gDNA was digested by HindIII restriction enzyme (New England Biolabs, Ipswich, MA USA), and the digested DNA was subjected to ligate by using a T4 ligase (Roche, Basel, Switzerland). The ligated DNA sample was used as a template DNA for PCR. The primers are Tn-F (5′-TAACAGGTTGGCTGATAAGTCC-3) and Tn-R (5′-GCATGCCTGCAGGTCGACGG-3). The amplified PCR bands were gel purified and sequences were determined by a Sanger sequencer.

## 3. Results and Discussions

The purpose of this study is to conduct genetic and physiological studies by constructing a transposon mutant library in the *P. antarctica* PAMC 27494 strain. For the construction of the transposon mutant library, we used conjugation methods. *E. coli* SM10(λpir) pBTK30 and *P. antarctica* PAMC 27494 strains were used as donor and recipient strains, respectively. The pBTK30 plasmid encodes the Mariner transposon-containing gentamicin-resistant gene, and it has been used in *Pseudomonas aeruginosa* [26]. In this study, we judged whether this approach can be applied in *P. antarctica*, and if so, we optimized the protocol for *P. antarctica* since the optimal growth temperature and antibiotic resistance are different from that of *P. aeruginosa*.

SM10(λpir) pBTK30 and PAMC 27494 strains were cultured in LB medium at 37 °C and 20 °C, respectively. Further, 50 μg/mL of ampicillin was supplemented to maintain a plasmid of the donor strain. Overnight cultured cells were harvested by centrifugation at 7000× *g* for 2 min at room temperature, and subsequently, the optical density at 600 nm (OD_600_) was adjusted as 10 by resuspending with an appropriate volume of LB medium. The adjusted donor and recipient cells were mixed in a 1:1 ratio of 500 μL each, and 50 μL of cell mixtures were spotted ten times on two LB–agar plates. For the negative control, only donor cells or only recipient cells were spotted on LB–agar plates. To determine the optimized conjugation condition, mating was conducted by varying the temperature, time, and ratio of donor and recipient cells. The results are discussed in the following paragraphs. After mating, the spots were scraped and resuspended in 1 mL of VBMM medium as the selection medium [27]. The slurry of cells was spread on a VBMM–agar medium containing 0.2% citrate acid and gentamicin. Citrate acid was added as a sole carbon source that cannot be utilized by *E. coli*, resulting in the inhibition of the growth of donor cells [29]. Upon the transfer of the transposon from pBTK30, a gentamicin resistance gene cassette was inserted into the chromosome of the recipient *P. antarctica*, and it survived in selection plates, as pBTK30 could not be replicated in *Pseudomonas* [26]. Then, 200 or 600 μL of cell slurries were applied for spreading to selection plates with a diameter of 9 and 15 cm, respectively. The plates were incubated at 20 °C for four days.

We optimized the experimental conditions to improve the efficiency of conjugation in *P. antarctica* (Figure 1A). It was reported that *P. antarctica* showed the optimal growth rate at 20–30 °C [7,15]. Therefore, conjugation should be carried out at a proper temperature for *P. antarctica*, which is different from the condition for *P. aeruginosa* [26]. We tried various temperatures for conjugation to determine the optimal condition for *P. antarctica*. The mating efficiency according to the ratio of donor and recipient cells was also tested. In addition, since *P. antarctica* has not been well characterized for its resistance to gentamicin, various gentamicin concentration ranges from 100 to 400 μg/mL were tested in the selection medium.

The optimal gentamicin concentration was determined to be 200 μg/mL, because some colonies of the *P. antarctica* PAMC 27494 strain appeared at less than 200 μg/mL of gentamicin (Figure 1B,C, Appendix A). Mating between donor and recipient cells was performed at two temperatures, 20 °C and 30 °C, and mating at 30 °C yielded more transconjugants than mating at 20 °C (Figure 1C). The incubation of selection plates for up to four days at 20 °C was recommended, as spontaneous gentamicin-resistant mutants appeared after more than four days of incubation. The largest number of transconjugants was produced when the ratio between the donor and recipient cells was 1:1 (Figure 1B,C). The cell density of donor or recipient cells should not exceed OD_600_ 40 and 20, respectively. Otherwise, it becomes difficult to secure independent colonies due to the excessive amount of cells on the plate. In addition, we tried various conjugation times, for 3, 8, and 24 h. The conjugation times longer than three hours tend to generate gentamicin-resistant colonies on the negative control plates. Therefore, we concluded that the optimized condition to efficiently obtain colonies of transconjugants is three hours of mating at 30 °C with a 1:1 ratio of donor and recipient cells. Further, 200 μg/mL of gentamicin is needed to supplement the selection medium and the incubation of selection plates should not exceed more than four days at 20 °C.

To generate a transposon insertion library with a size of over 20,000 transconjugant colonies, approximately 8 mL of the mating mixture prepared by the optimized protocol is needed for spreading on selection plates. After four days of incubation, colonies were scraped and adjusted as OD_600_ 10 by using VBMM–gentamicin medium containing 16% glycerol to store at −80 °C.

Finally, we verified the insertion of transposon in the transconjugant, and also identified the transposon insertion site in the genome. Firstly, the acquired gentamicin resistance was checked by streaking on the medium containing gentamicin (Figure 2A). The isolated transconjugant strain exhibits gentamicin resistance in both LB and VBMM medium, but the parental strain showed sensitivity to gentamicin as a negative control. In addition, the integration of the transposon could be confirmed by PCR using a pair of primers that bind specifically to the transposon sequences (Figure 2B). To identify the transposon insertion site, inverse PCR and subsequent sequencing were performed as described in Materials and Methods [28]. This experimental result will provide the sequence of the transposon and also the flanking region of the transposon insertion site. The transposon insertion site of the tested transconjugant was identified at 3,549,551 bp, which corresponds to the intergenic region between RS15985 and RS15990 genes (Figure 2C, Appendix A). Therefore, we were able to verify the insertion of transposons and also identified the transposon insertion site in *P. antarctica*.

The next-generation sequencing method can be coupled with transposon-based genetic screening, which is called transposon sequencing (Tn-Seq). This method determines transposon insertion sites as well as transposon insertion frequency to find the genes with altered transposon insertion frequency after selection [30]. If certain genes are required for survival against the condition of selection, the transposon insertion frequency will be decreased. In contrast, for genes exhibiting a disadvantageous effect on survival against the selection condition, the transposon insertion frequency will be increased. Therefore, Tn-Seq is a powerful and high-throughput method to find both advantageous and disadvantageous genes at the selection condition.

In summary, we established the optimized transposon insertion protocol for *P. antarctica*, which is lacking in genetic studies. The isolation of the transposon mutant under various temperatures could reveal the secrets of the survival strategies with which the bacteria have adapted to low temperatures. In addition, the genes involved in the production of antimicrobial substances could be also discovered by using the transposon mutant library of *P. antarctica*. For the high-throughput screen, the Tn-Seq method can be applied.

## Figures and Tables

**Figure 1 microorganisms-11-00118-f001:**
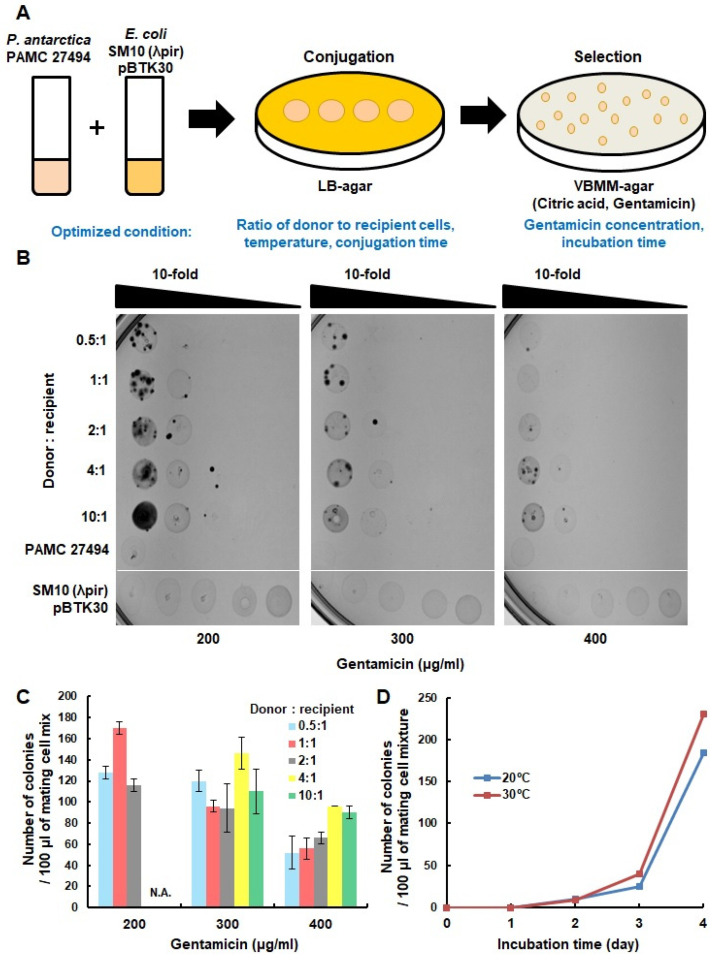
Optimization of transposon insertion methods in the *P. antarctica* PAMC 27494 strain. (**A**) Schematic workflow of transposon mutagenesis is shown. Donor and recipient cells were SM10(λpir) pBTK30 and *P. antarctica* PAMC 27494, respectively. The optimized parameters are stated. (**B**) A total of 10 μL of a 10-fold dilution of mating cell mixtures with various ratios of donor to recipient cells were spotted on VBMM–agar medium containing gentamicin. As a negative control, only donor cells or only recipient cells were spotted on the plate. Only the SM10(λpir) pBTK30 strain was spotted according to the applied density (ordering from left to right). (**C**) The colonies on each plate were counted, and the average was converted to 100 μL. The graph shows the efficiency according to the ratio of donor cells and recipient cells. Samples in which it was not possible to count the colonies due to background cells are indicated as ‘Not Applicable (N.A.)’. (**D**) The efficiency of conjugation at 20 °C and 30 °C for three hours of mating was compared.

**Figure 2 microorganisms-11-00118-f002:**
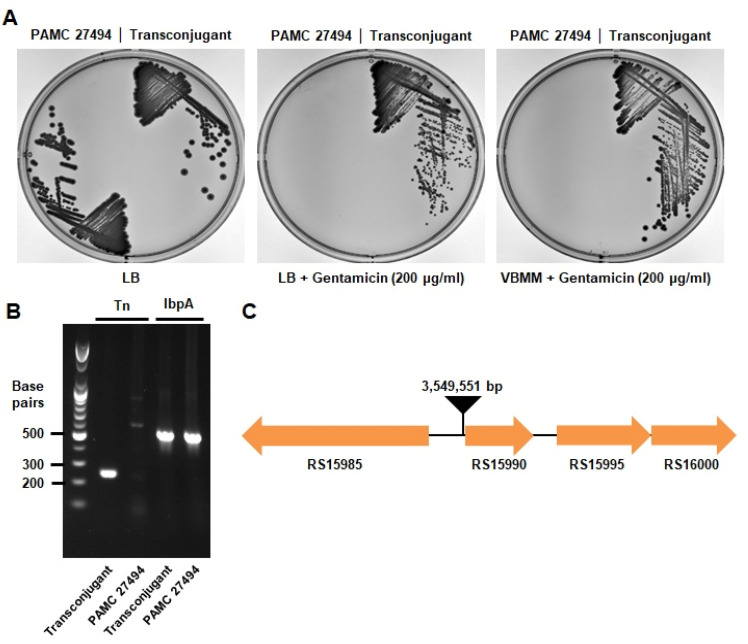
Confirmation of transposon insertion in the *P. antarctica* PAMC 27494 strain. (**A**) The *P. antarctica* PAMC 27494 strain and its transconjugant were streaked on LB, LB–gentamicin (200 μg/mL), and VBMM–gentamicin (200 μg/mL) and incubated at 20 °C for four days. (**B**) To verify the insertion of the transposon, PCR was performed by using two primer sets for targeting transposon-specific sequences and the *ibpA* gene as a PCR control. (**C**) By performing inverse PCR and subsequent sequencing, the insertion site of the transposon was determined, which is at 3,549,551 bp, the intergenic region between RS15985 and RS15990 genes in *P. antarctica*.

## Data Availability

The data presented in this study are available on request from the corresponding author.

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
