# Peer review of "Optimization of Transposon Mutagenesis Methods in Pseudomonas antarctica"

_microorganisms, 2023, doi:10.3390/microorganisms11010118_

Round 1

Reviewer 1 Report (Previous Reviewer 3)

The revised manuscript is well improved. The paper should be accepted. 

Reviewer 2 Report (Previous Reviewer 2)

I have reviewed the authors' changes, and I believe that the modified

manuscript is suitable for publication. Thank you to the authors for

their thorough consideration of my comments. 

This manuscript is a resubmission of an earlier submission. The following is a list of the peer review reports and author responses from that submission.

Round 1

Reviewer 1 Report

The manuscript by Kim et al, “Generation of transposon mutant library for Pseudomonas antarctica”. In this study authors aimed to develop a conjugation-based method using a transposon mutagenesis tool to identify genes of interest in Pseudomonas Antarctica.  

Overall, I did not find anything novel in this manuscript beside the organism. This article is more suitable for methods papers or protocols. 

Reviewer 2 Report

Peer review for Kim & Li“Generation of transposon mutant library for Pseudomonas antarctica

Summary:

Transposon mutagenesis is a widely used technique to probe gene function, and in this study Kim & Li applied a conjugation method to insert mariner type transposons throughout the genome of P. antarctica PAMC 27494. Successful insertion of the transposon is demonstrated by PCR and sequencing, but full characterization of the library was not performed. While this study contributes to the field by demonstrating the first use of transposon mutagenesis in P. antarctica, it would benefit from the following modifications before it is fit for publication in Microorganisms:

Major comments:

1.      The title and the abstract/introduction describe this study as the “generation of a transposon mutant library”, but the data presented are not sufficient to support this claim. Next-generation sequencing is required to demonstrate that transposons integrated throughout the P. antarctica genome (i.e., in most protein coding genes), but this was not done, so the authors cannot claim that transposon insertion coverage was sufficient to use these mutants for fitness experiments. The authors have only examined a single insertion by Sanger sequencing, and only two agar plates were used for conjugation, so there is not enough detail presented to claim that a genuine library has been created. The title should be changed to “Optimization of transposon mutagenesis methods in Pseudomonas antarctica” or something similar, and claims about library generation should be removed from the text.

2.      Some modifications to the introduction would improve the framing of the study:

a.      The first paragraph contains too much detail about the Pseudomonas genus. One or two sentences about the diversity and properties of pseudomonads should be sufficient.

b.      The rest of the first paragraph could then be devoted to discussion of other transposon libraries in pseduomonads and/or the merits of genomic characterization in extremophiles (e.g., P. antarctica).

c.      The introduction should also describe some fundamental information about the metabolism of P. antarctica since this is fundamental to its characterization with transposon mutant libraries. Is P. antarctica easily cultured in the laboratory? What are some hallmarks of its metabolism?

d.      The end of the introduction would benefit from some discussion of how a transposon mutant library in P. antarctica would be applied (for gene/pathway discovery, etc.).

3.      Line 49: “Transposons can integrate into the genome at a random position…” This is incorrect. Only some transposon types can integrate randomly. Many of them (e.g., Tn7) target a specific genomic locus.

4.      Line 92: the ibpA gene was used as a control for PCR, but is it possible that this gene could have a Tc1 insertion? If not, it should be stated that this is an essential gene.

5.      Lines 126-134: Beginning at “After mating, the spots were scraped…”, these sentences should be moved to the Methods section.

6.      Lines 149-150: “…spontaneous gentamicin-resistant mutants may appear after more than four days of incubation.” No data is provided to support this claim. How was this determined? Were mutants sequenced or PCR amplified to show that they lacked the gentamicin resistance gene from pBTK30?

7.      Figure 1: This should contain three sub-figure labels. A = spot plate images, B = number of colonies as a function of incubation time, C = number of colonies as a function of gentamicin concentration. References in the text should be updated accordingly.

8.      Lines 180-187: The Sanger sequencing read for the single analyzed transposon insertion site should be provided in the supplementary information, ideally as a file format that includes the sequencing peaks.

Minor comments:

1.      Throughout the paper, abbreviations of bacterial species are inconsistent. Pseudomonas antarctica only needs to be spelled out once in the introduction. Every subsequent mention can be written as “P. antarctica”.

2.      Despite the English language editing service, several grammatical and spelling errors remain:

a.      Line 35: “genetic studies have not actively done” – should be corrected to “genetic studies have not actively been done”.

b.      Line 57: “..genes are required to survival”  - should be corrected to “genes are required for survival”.

c.      Line 83: “flacking” region should be corrected to “flanking” region.

3.      Line 71: it is unclear how colonies were handled after conjugation and selection. Were they scraped into selective media? Was any outgrowth performed? Were aliquots of the transconjugants frozen/stored?

4.      Microbial species names, e.g. Pseudomonas aeruginosa, should be highlighted in the References.

Reviewer 3 Report

Transposon mutagenesis is a useful tool to screen genes of interests. This paper describes the method of generation transposon insertion mutants using transfer of conjugational plasmid encoding a transposon. This paper is quite interesting and should be published.